# Highly Resolved Runoff Path Simulation Based on Urban Surface Landscape Layout for Sub-Catchment Scale

**Tian Bai** [1] , **Klaudia Borowiak** [2] , **Yawen Wu** [1,*] **and Jingli Zhang** [1,*]

1. College of Horticulture and Landscape, Yunnan Agricultural University, Kunming 650201, China; tbai@ynau.edu.cn
2. Department of Ecology and Environmental Protection, Poznan University of Life Sciences, Piatkowska 94C, 60-649 Poznan, Poland; klaudia.borowiak@puls.edu.pl
* Correspondence: yawenwu@ynau.edu.cn (Y.W.); jinglizhang@ynau.edu.cn (J.Z.); Tel.: +86-199-8762-6779 (Y.W.); +86-138-8828-0046 (J.Z.)

**Abstract:** The present study explored the regularities of the path and network structure of surface runoff formed under the influence of urban surface landscapes. We used unmanned aerial vehicle sensors to examine terrain and land use/cover change. The sub-catchments of a typical city, Luohe, China, were evaluated for the effect of landscape on surface runoff. Landscape and topographic parameters from 166 urban sub-catchments in Luohe were obtained by measuring digital surface models and orthophoto maps. The minimum cumulative resistance model was used to simulate potential runoff and 491,820 potential runoff paths, connected upstream and downstream, were obtained in 166 sub-catchments. The chi-square test was used to compare simulation runoff paths and actual runoff depth, with the results showing that they led to the same distribution trend. When the gravity coefficient was greater than 18.93, path disconnection occurred among 166 sub-catchments, with a decrease in channels. The potential runoff distribution appeared in aggregation; as the gravity coefficient increased from low to high, aggregation showed a trend of increasing initially but subsequently decreasing. The initial runoff formed sub-catchments with high gravity coefficients, then accumulated and spread to the others. It is important that proper measures are taken to establish a unified planning of the city's surface landscape in order to produce suitable surface runoff distribution.

**Keywords:** urban surface runoff; risk assessment; distribution regularities; sub-catchment management

## 1. Introduction

Rapid urbanization has caused huge changes in urban sub-catchments and profoundly affected surface runoff. It not only leads to a decrease in infiltration, but also increases runoff and ultimately affects the hydrological cycle and urban environment [1,2]. Studies have confirmed that land use/cover change (LUCC) induced by urban expansion and the subsequent increase in the areas of impermeable ground surface are considered the main causes of changes in urban hydrology. In addition, the expansion of impermeable areas is linked to urban disasters, such as flash floods, non-point source pollution, water shortages, and river deterioration, as well as climatic change [3,4]. Studies have shown that without factoring in the potential impacts of climate change, cities around the world will be 2.7 times more likely to suffer flood hazards by 2030. For example, a devastating flash flood occurred in Beijing, the capital of China, on 21 July 2012. This flooding event resulted in a loss of 79 lives and USD $1.67 \times 10^{12}$ in loss. The connection between surface runoff and landscape has been steadily established through further research. Impervious landscapes contribute to increased accumulation of stormwater and urban floods [5], and reducing impervious surfaces and planting scattered vegetation that can maintain and purify water quality and quantity [6]. Impervious areas amplify the spread of surface runoff into urban areas. The common conventional solution is often to discharge water directly through channels and

pipes [7]. Thus, management advice often focuses on limiting imperviousness, especially in and adjacent to the riparian zones of existing surface water bodies. Conventional drainage systems, used to direct stormwater to streams, rivers, retention ponds, and reservoirs, are also based on conventional municipal infrastructure management modes [8]. However, in China, this approach has been shown to be inadequate in dealing with many recent intense rainfall events. Continuous urban expansion and frequent urban renewal projects lead to the deterioration of urban ecological structures and functions, which has resulted in an increase of urban flood frequency from a sudden increase in surface runoff during heavy rain. In 2016, for example, 1508 counties in 28 provinces (cities and districts) in China were hit by pluvial urban waterlogging [9].

Currently, green space has been found to be an important tool for managing stormwater and, in turn, treating green infrastructure (GI) as an important part of urban management systems is becoming more frequent [10]. Some of the popular strategies are low impact development (LID), best management practices (BMPs), smart growth (SG), and water-sensitive urban design (WSUD) [8,11]. However, to date, sustainable stormwater management implementation has encountered barriers and challenges due to rapid urbanization. Principles, legal acts, and practices in the overall development of China are still at their initial stage. Most current studies are focused on increasing rainwater infiltration by improving surface permeability. However, one serious problem is ignored—due to very high economic cost, does all the land need improved permeable area? Changing the infiltration capacity of all urban land requires urban renewal and long-term construction. While in the implementation process, cities are still experiencing severe flooding events due to rapid urbanization. The complexity of urban development, which changes urban land use and city structure, makes it difficult to identify the source, path, and term of the overall surface runoff of the city. In the past, city roads and communities in China's second- and third-tier cities were usually considered as independent units. The curbs isolate the green spaces from the roads. The roads are mainly responsible for the direction of runoff, especially due to changes in the inclined angle of the local area. However, such approaches fail to estimate the relationship between upstream and downstream runoff, so that the excess surface runoff would flow into the downstream region and result in an increased probability of floods in the downstream region. Therefore, we need a more comprehensive method to identify the urban surface runoff distribution path in the city to increase the efficiency of sustainable surface runoff management and improve the economy of design and construction; new strategies need to be developed for the effective extraction of information.

The purpose of the present study was to explore the regularities of the path and network structure of surface runoff formed under the influence of urban surface landscapes through the use of unmanned aerial vehicle (UAV) sensors (including terrain and LUCC). To effectively minimize the negative impacts of surface runoff, we identified the surface landscape, calculated the source–sink process of surface runoff, extracted the characteristic information, and implemented surface runoff management according to the feature information [12]. We hypothesized that some factors, such as the variability of the surface landscape types, may lead to differences in the resistance of the urban surface runoff distribution process. Therefore, these differences in surface landscape will cause changes in the flow and direction of surface runoff, as well as in different distributions. Through simulating the distribution of potential runoff, the key areas to be transformed, the inlet and outlet of green space, can be identified. Diversion design can also be carried out at the intersection of surface runoff [13].

## 2. Materials and Methods

Geographic information systems (GIS) and remote sensing (RS) were used. These technologies can provide the necessary tools to tackle the problems with up-to-date spatial information [14]. RS of surface landscape layout of spatial distribution surface runoff trend, a rather new topic for landscape or urban planning, was employed. This technical develop-

ment is based on the application of UAVs to acquire images of high spatial resolution. UAV sensors can acquire surface landscape data which is needed to model the urban surface runoff within a city for analysis. The strength of our analysis is that we acquired a 0.09 m resolution orthogonal projection and 0.09 m vertical resolution digital surface model (DSM) of the entire urban study area through the UAV [15].

### 2.1. Study Area

Luohe is located in middle Henan Province, China (Figure 1). The administrative division has an area of 2617 km$^2$, with 78 km$^2$ dedicated to the city with a population of $2.84 \times 10^7$, where approximately half of the people lived in the central district at the end of 2018. Our study focused on this central district of 169.38 km$^2$, where the urbanization rate is 52.47%. The construction area of Luohe increased from 24 km$^2$ in 1999 to 83 km$^2$ in 2018, with an annual expansion rate of 3.11 km$^2$/year. Green space accounts for 39.8%, with 12.8 m$^2$ of green space per capita of the city. While the green space increased from 37.4% in 1999 to 41.3% in 2019, this increase was due to the expansion of the city area rather than increased green space within the city itself, and it distribution still remains imbalanced [16].

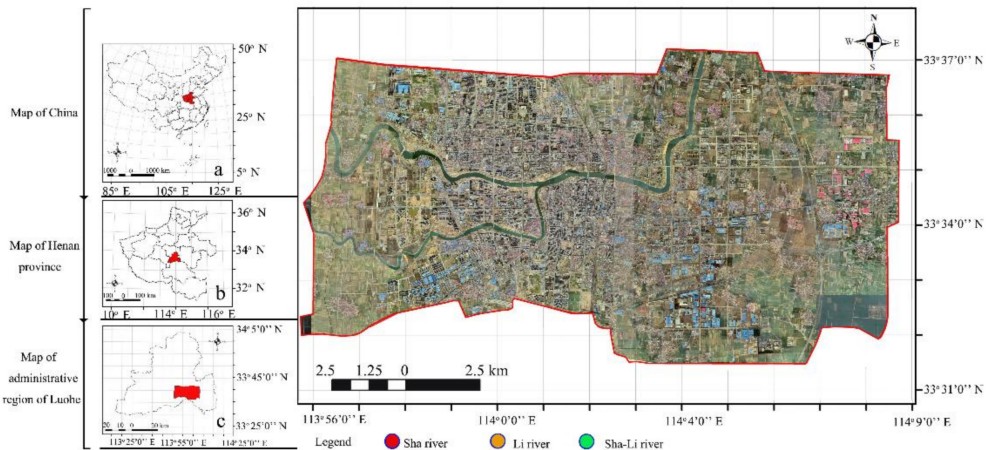

**Figure 1.** Geographical location map of Luohe research area.

The city is an alluvial plain by construction of embankments along the confluence of the Sha-li River, increasing flood discharge capacity to protect the city from most river-cresting events. However, the decrease of permeable surfaces has led to an increment of urban waterlogging events, in terms of increased frequency and severity inside the city. Landscape layout is one of the important factors that affect urban waterlogging.

### 2.2. Land Use and Cover Classification

We acquired a digital land use/land cover map using an aerial photography image from a Trimble UX5, fixed-wing, unmanned aerial vehicle (UAV), with a 48 megapixels, silent frame and 0.09 m ground resolution, using a 25 mm focal length lens. The total extent of the image was 169.38 km$^2$. Details of the UAV flight included: flight altitude 300 m, a single flight covering an area of 3.7 km$^2$ flight mode for the 'Z'-type lane, 5:1 aspect ratio of a rectangle, and 80% image overlap rate. The post-processed kinematic and global navigation satellite technology in the UAV system established pinpoint image locations (accuracy > 0.15 m). To obtain a clear land classification and patch boundary identification, aerophotography was conducted during the defoliating period. We converted raster images into vector layers for classification, which were interpreted and verified manually using updated planning maps. This work was based on ArcGIS 10.4. Surface landscape factors were allocated into two groups—permeable (group 1) and impermeable (group 2) — which were categorized into seven classes and classified by dominant human land use based on their influence on surface runoff.

The permeable region included landscaping space (#P1: forest, woodlots, green space, lawns), water areas (#P2), agricultural land (#P3: cropland, plantation, pasture), and unused land (#P4: bare soil, construction sites, no vegetation). The impermeable region included streets and roads (#I1: highway, pavement), roofs (#I2), and locus publicus (#I3: commercial plazas, park squares, paved parking lots). Although construction sites (35% of P4) often impedes permeability due to soil compaction, we included this class in the permeable region since the disturbed soil surface was rough and permeable, at least before weathering and subsequent sealing of the soil's surface (Figure 2).

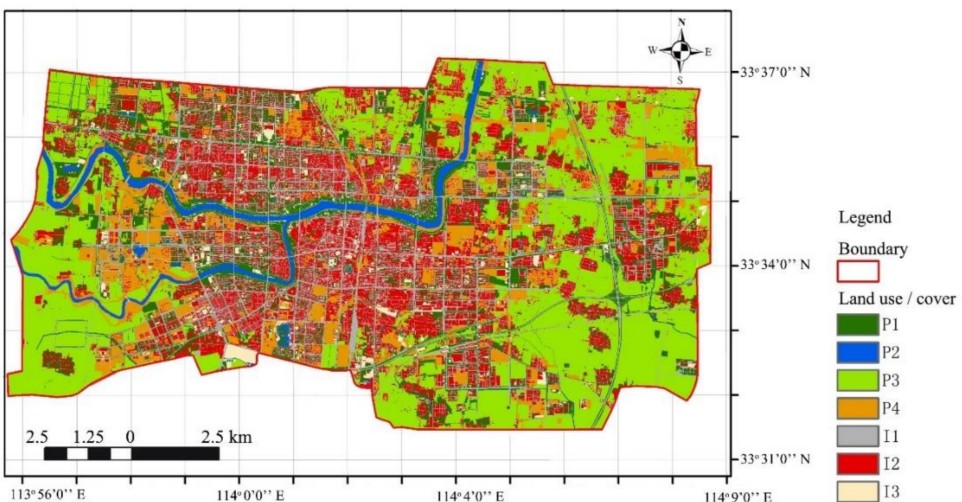

**Figure 2.** Land use/cover classification of Luohe central district with divisions based on permeability.

### 2.3. Generation of a High-Resolution Digital Surface Model

In urban GIS and management, digital surface model (DSMs) are used for generating object models. A DSM was generated to obtain the inverted elevations and slopes of the stormwater structures. The preliminary step was to segment the interesting areas. We acquired a 0.09 m vertical resolution DSM of the entire urban study area through the UAV. Then, the terrains could be extracted from DSM (Figure 3). Since urban appearance is complicated with buildings, green space, water, etc., terrain extraction was implemented through several stages. In the first stage, building roofs were extracted from the DSM. In the second stage, marker-based sub-catchments were implemented to get the boundaries of the objects above the ground [17]. The result of marker-based buildings and sub-catchments were merged to improve the urban terrain extraction accuracy.

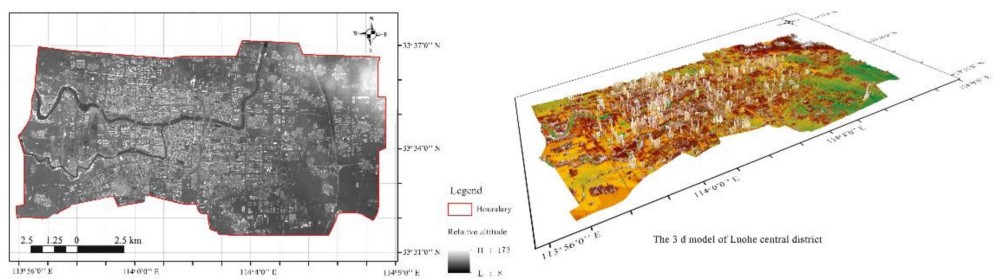

**Figure 3.** Terrain analysis based on DSM of Luohe central district.

### 2.4. Sub-Catchment Division

Sub-catchments were defined according to physical parameters, forming units of urban surface runoff [18,19]. Usually, sub-catchments are divided based on outlets to the sewer system, DSMs, and the urban pipe network in the urban central district within the clear flow direction [20]. Therefore, we acquired 166 sub-catchments. There was a

high density of buildings and a network of roads, so the sub-catchments were small and in large numbers. However, the areas at the edge of the city were larger. Among the 166 sub-catchments we analyzed, the largest was 11.02 km$^2$, the smallest was 0.072 km$^2$, and the average area was 1.03 km$^2$ (Figure 4).

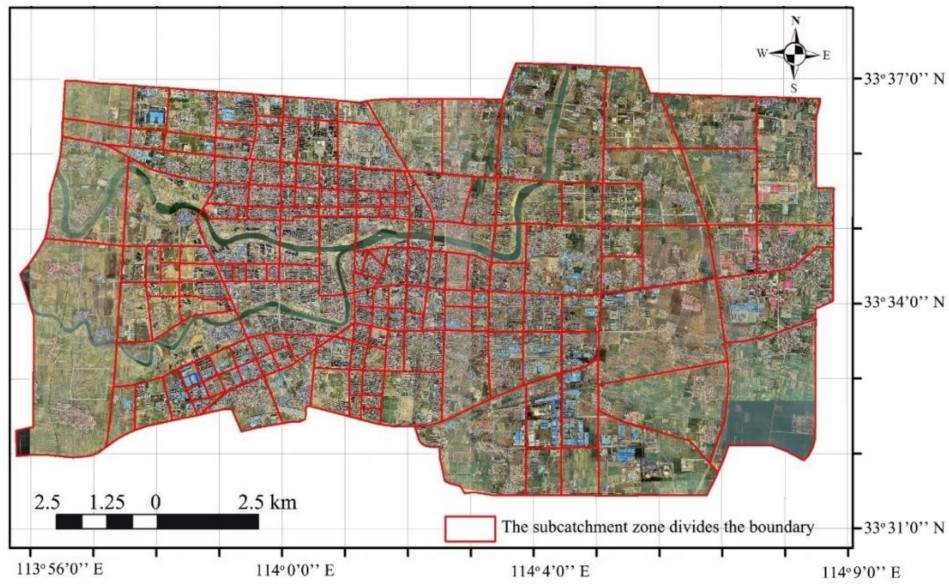

**Figure 4.** Sub-catchments division of Luohe central district.

### 2.5. Minimum Cumulative Resistance (MCR) Model

With respect to the distribution of runoff process, an urban landscape layout could be divided into a "source" or "sink" [21]. The source consists of those processes in the upstream area from which surface runoff is produced, and the sink consists of the downstream area that receives stormwater runoff [22]. Although source and sink of landscapes have opposite qualities, a sink in one process may become a source in another. The main purpose of the proposed the source and sink landscape theory was to explore how the status quo of the urban landscape would affect the distribution of the runoff process. In turn, the runoff process would help in the identification of a suitable spatial pattern and overall urban landscape in every region [20]. The transformation between source and sink occurs as a mutual conversion process for the path of surface runoff, which was achieved by overcoming various forces of resistance. They were then integrated to reflect the resistance surface [23].

The MCR model essentially reflects the degree of resistance in landscape layout. The cost distance in the MCR model is different from the actual distance. Instead, it emphasizes the relative spatial relationship between points. [24]. This cost distance is based on calculating the resistance coefficient when the target path passes through different landscape units [25,26]. Based on the resistance value of MCR when a target path passes through a specific unit, we can determine the connectivity between the two points. Usually, the "source" is the initiator of runoff distribution and flows into the "sink" through different landscape units. To determine the minimum cumulative resistance value, MCR, the following modified formula is:

$$MCR= f_{\min} \sum_{j=n}^{i=m} \left( D_{ij} \times R_i \right) \qquad (1)$$

where f is a function of the positive correlation that reflects the relation of the least resistance from any point to any point in space and the characteristics of the landscape base surface, min is the minimum value of cumulative resistance produced in any processes of source unit $i$ transforming into a different sink unit $j$, $D_{ij}$ is the spatial distance between unit $i$ and

source unit $j$, and $R_i$ denotes the resistance coefficient that exists in transition from source unit, $i$, to sink unit, $j$.

$R_i$ can be determined by the value of $i$, and the path from unit $i$ to unit $j$ produces different resistance values. When $i$ has been determined, calculating $MCR$ requires selecting the path with the least resistance value of the spatial distance.

In this paper, source refers to the landscape units that initiate the process by which surface runoff occurs. Sink refers to the landscape units that receive inflow from the source runoff. During MCR calculation, the input of the source and sink can be a patch or a combination of patches, and sources could be connected or unconnected in space. We assumed that the divided sub-catchments were the sources and sinks of each other. Surface runoff has the common characteristics of liquid flow and can flow between different sub-catchments [27].

### 2.6. Chi-Square Test of Simulation Results

The chi-square test is a widely used hypothesis testing method, which is used to calculate the degree of fitting between the observed sample and the simulated value [28]. The chi-square test was used to measure the relationship between simulation value and actual runoff depth. The runoff depth values were, respectively, the runoff depth on 27 June, 22 July, and 30 August 2020 at the 36 flooding points in Luohe [16]. Simulation values were potential runoff paths from the MCR model.

### 2.7. Gravity Model

Gravity has been one of the most successful empirical models in spatial analysis. Interactions between spaces, such as information flow and material flow have been related to size and distance. In urban network structures, interactions between objects such as cities or areas lead to models remembering Newton's gravity law, where the sizes of the objects play the role of mass [29]. Incorporating deeper theoretical foundations of gravity into recent practice lead to a richer and more accurate estimation and interpretation of the spatial relations described by gravity. Wider acceptance has followed, and suggestions are made for promising future research [30].

$$G_{ab} = \frac{N_a N_B}{D_{ab}^2} = \frac{\left[\frac{1}{P_a} \times \ln(S_b)\right]\left[\frac{1}{P_b} \times \ln(S_b)\right]}{\left[\frac{L_{ab}}{L_{max}}\right]^2} = \frac{L_{max}^2 \ln(S_a)\ln(S_b)}{L_{ab}^2 P_a P_b} \tag{2}$$

where $G_{ab}$ is the interaction between surface runoff paths $a$ and $b$ of the sub-catchments, $N_a$ and $N_b$ are the weights between $a$ and $b$, $D_{ab}$ is the standardized value of potential runoff path resistance in $a$ and $b$, $P_a$ is the resistance of sub-catchment a, $P_b$ is the resistance of sub-catchment b, $S_a$ is the area of sub-catchment $a$, $S_b$ is the area of sub-catchment $b$, $L_{ab}$ is the accumulated resistance value of the surface runoff path from $a$ to $b$ in the sub-catchment, and $L_{max}$ is the maximum resistance value of the potential surface runoff path in the entire study area [30].

### 2.8. Spatial Autocorrelation

When the values of variables observed at spatial positions resemble each other more than expected for a randomness model, the variable is said to be spatially autocorrelated [31]. It is a measure of the degree to which objects, situated in close proximity, have a tendency for similar values of a given index. In studies of surface runoff, spatial autocorrelation may arise from the underlying causal variables of landscape layout and various other factors [32]. The spatial distribution of runoff formation can be revealed by analyzing runoff density in each sub-catchment. The potential runoff density is $d = l/a$, where d is the potential runoff density in each sub-catchment, l is the length of the potential runoff, and $a$ is the area of each sub-catchment.

Moran's *I* is one of the most popular indices for assessing spatial autocorrelation in spatial data, and is defined as follows:

$$I = \frac{m}{\sum \sum w_{ij}} \times \frac{\sum\limits_{i}^{n} \sum\limits_{j}^{n} w_{ij} Z_i Z_j}{\sum\limits_{i}^{n} Z_i^2} \qquad (3)$$

In expression (3), $z_i = p_i - p$, where *pi* is the runoff density in the *i*th units, for $i = 1..., m$, and $W_{ij}$ is a weight assigned to every pair of units. Conventionally, $w_{ii} = 0$. This weight indicates the importance of the locality pair in measuring the spatial autocorrelation. Usually, units that are adjacent or closer together than some distance thresholds have weights assigned to unity, with the other weights being zero. Other weight structures are also possible. Moran's I usually ranges between 1 and −1, with positive values indicating similarity between geographic neighbors, negative values indicating dissimilarity, and values close to 0 indicating a random pattern. In recent years, spatial autocorrelation has come into its own in studies on urban rainwater distribution.

Industrial zones, business zones and high-density residential zones tend to have a greater influence on runoff [33,34]. Chang et al. [35] examined the spatial patterns of annual runoff ratios and their variability and identified the determinants of runoff indices for 238 reference basins with low levels of anthropogenic influence and 1352 non-reference basins with substantial levels of anthropogenic influence variability across the contiguous U.S. The research results showed local specific relationships between runoff indices and landscape factors [35]. The geo-statistical properties of soil moisture patterns from five different locations in Australia and New Zealand have been researched. The results showed that the processes controlling spatial patterns can change between temporal and spatial scale with catchment moisture status; however, when similar general phenomenon reoccur in a catchment, similar spatial patterns result [36].

## 3. Results

### 3.1. Identification and Analysis of Urban Surfaces

In the course of rainfall, surface runoff mainly spreads in the horizontal direction, and the horizontal movement mechanism is the method of redistributing surface runoff in the process of rainfall. It is also the basic form of the catchment confluence process.

In the process of horizontal confluence, surface runoff will be influenced by the surface landscape, including terrain factors (altitude, slope, relief amplitude, and roughness) and LUCC, which produce resistance along the way and local resistance, making the runoff continually change direction to form a network. The network path of surface runoff is determined by the quality of the source and sink, the resistance between the source and sink, and the distance between the source and sink. Among them, the surface landscape plays an important role in the flow process of surface runoff. Combined with literature materials, this study evaluates the resistance of different surfaces generated along the path and local resistance and obtains the confluence resistance consumption surface [27].

First, we extracted the single factor cost surface through satellite imaging and the DSM. Then, values of 120–1000 were assigned to different indices according to different factors (Table 1) and the AHP method was used to calculate the weight base on references (Table 2) [37–39]. Finally, different single factor cost surfaces were superimposed by raster calculation to form the comprehensive cost surface [40,41] (Figure 5).

### 3.2. Distribution Characteristics of Potential Surface Runoff Paths

The potential runoff paths constructed by the MCR model revealed 491,820 runoff connections in total, distributed among 166 sub-catchments. The potential runoff paths were linear, and not only connected the upstream and downstream sub-catchments but also showed clear directions, pointing from sources to sinks (Figure 6).

**Table 1.** Resistance of surface runoff confluence process of Luohe.

| Resistance | 120 | 150 | 180 | 300 | 500 | 800 | 1000 |
|---|---|---|---|---|---|---|---|
| Terrain factors | | | | | | | |
| Altitude | 0–30 m | 30–50 m | 50–80 m | 80–100 m | 100–150 m | >150 m | |
| Slope | 0–5° | 5–10° | 10–15° | 15–25° | >25° | | |
| Relief amplitude | 0–15 | 15–30 | 30–60 | >60 | | | |
| Roughness | 0–12 | 12–24 | 24–36 | 36–48 | >48 | | |
| Landscape factor | | | | | | | |
| LUCC | Water | Road | Public Management Land | Green Space | Unused Land | Agricultural Land | Roofs |
| | P2 | I1 | I2 | P1 | P4 | P3 | I3 |

**Table 2.** Weight value on single factor of Surface Landscape Layout.

| Goal Layer A | Criterion Layer B | Weight | Index Layer C | Weight |
|---|---|---|---|---|
| Surface landscape layout | Terrain factors B1 | 0.75 | Altitude C1 | 0.14 |
| | | | Slope C2 | 0.35 |
| | | | Relief amplitude C3 | 0.19 |
| | | | Roughness C4 | 0.07 |
| | Landscape factor (LUCC) B2 | 0.25 | Green space C5 | 0.04 |
| | | | Water C6 | 0.07 |
| | | | Agricultural land C7 | 0.03 |
| | | | Unused land C8 | 0.01 |
| | | | Road C9 | 0.06 |
| | | | Roof C10 | 0.02 |
| | | | Public management land C11 | 0.02 |

$CI = 0.000$, pass consistency test.

The simulation results showed that 1.45% of the runoff intersected with the buildings but had an error rate of less than 5%. Causes of conjecture errors include the following: (1) only features and landscape elements were used as resistance characteristics for simulation in this study, (2) aerial imagery accuracy, and (3) the calculation of frictional resistance and local resistance to runoff is more applicable to the calculation of closed pipes or fixed transmission channels. The cost ratio index is used to quantify the average consumer cost of a network, and it mainly reflects the complexity of the network. The closer the cost ratio index is to 1, the more complex the network structure. The formula is as follows: cost ratio = 1 −L/d, (L is the quantity of runoff; D is the runoff length). The resulting cost ratio index was 0.71, and the network costs were relatively high.

Assumption was that potential paths were associated with depth of runoff. The depth of flooding points ($n = 36$) was measured (Figure 6) and Chi-square tests show that the signification ($p$) of repeat the experiment three times was greater than 0.05 (Table 3), respectively. That was to accept the null hypothesis, indicating that the overall efficiency of the simulated value of the null hypothesis was equal to the measured value. The difference between the potential paths and depth of runoff was not statistically significant, and there was no significant difference between the simulated value and the observed value, so the hypothesis of correlation was valid. Meanwhile, the verification result was consistent with my previous research tendency by Storm Water Management Model (SWMM) [16].

*3.3. Distribution Characteristics of Potential Surface Runoff Based on the Gravity Model*

The results of the gravity model showed that there was a great difference in the interaction intensity among sub-catchments. A gravity matrix was established to analyze the relationships among them (Table 4).

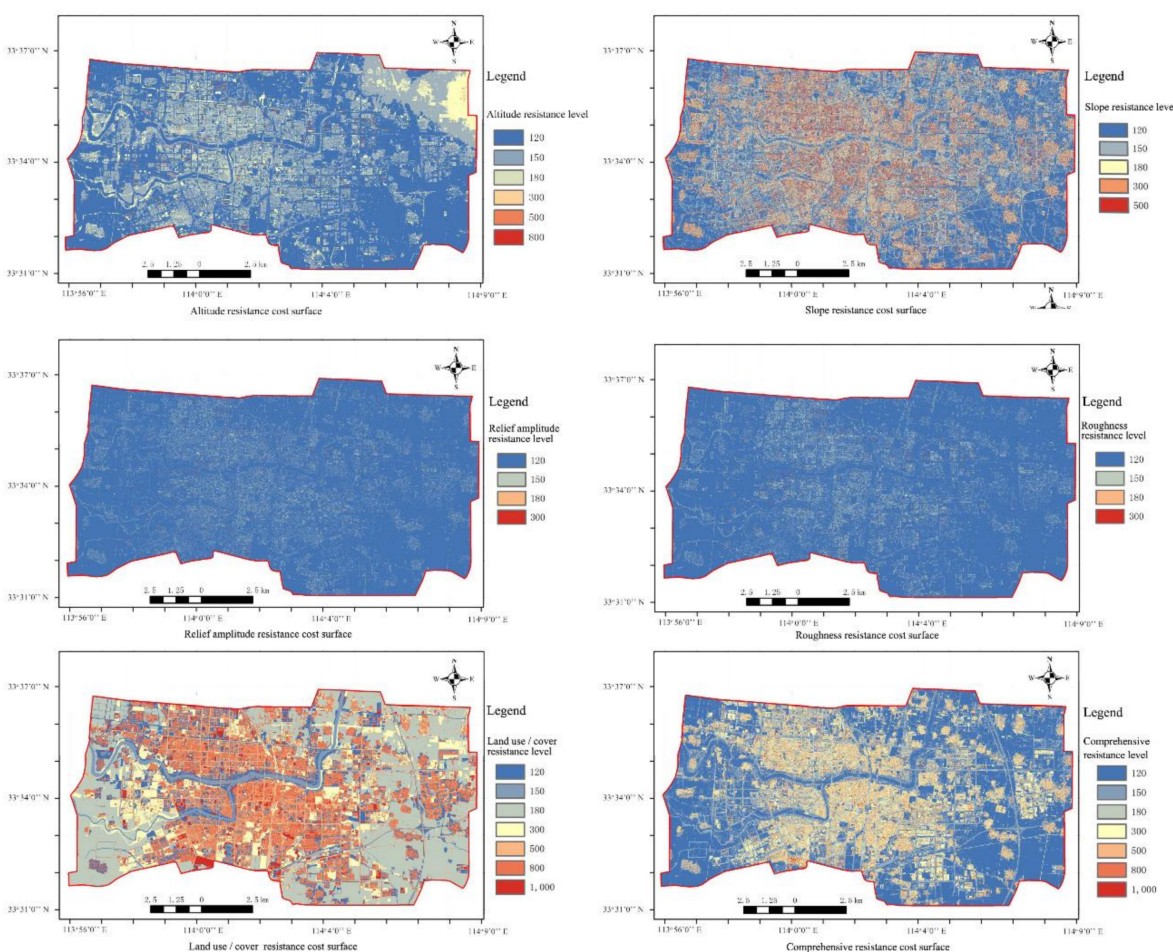

**Figure 5.** Surface runoff confluence process resistance surface of Luohe central district.

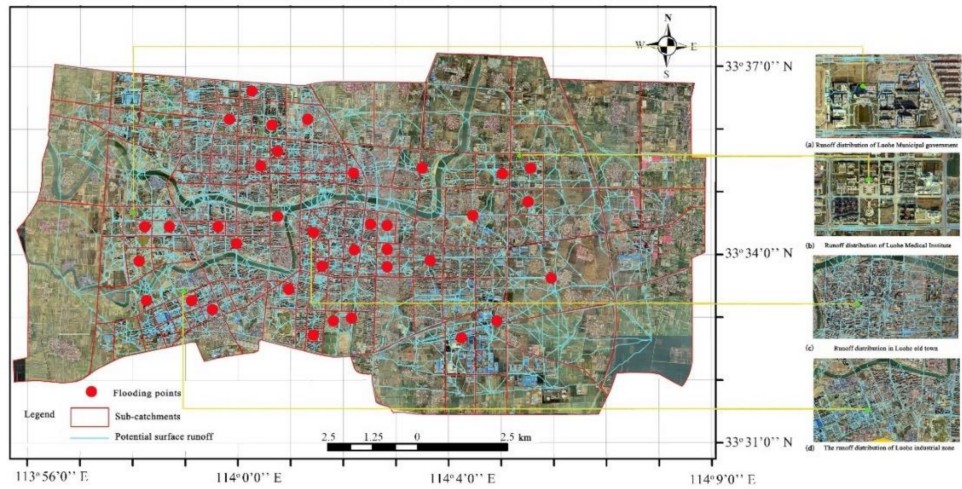

**Figure 6.** Distribution of potential runoff in of Luohe.

After screening, it was found that when the gravity coefficient was 18.93, the potential surface runoff network in 166 sub-catchments broke. Therefore, when the gravity coefficient was less than or equal to 18.93, all sub-catchments were connected. This was the maximum threshold of the connection of all the sub-catchments. Then, we selected gravity coefficient scenarios of 0, 18.93, 30, 40, 50, 100, 300, and 500 and calculated runoff under different gravity coefficient orders, namely, the gravity coefficient. The results showed that

the greater the sub-catchment connection strength between catchments, the more prone they were to surface runoff communication. From this outcome, we can infer the actual situations. When rainfall begins, the sub-catchments with the highest gravity coefficients generate runoff first, and with the increase in rainfall intensity, other sub-catchments generate runoff according to this parameter.

**Table 3.** Chi-square test of runoff depth and potential paths.

| Team | Sample Volume ($n$) | Chi-Square Value | Significance (Two-Sided Test) |
|------|------|------|------|
| 1 | 36 | 119.883 | 0.205 |
| 2 | 36 | 86.779 | 0.063 |
| 3 | 36 | 86.779 | 0.113 |

**Table 4.** Interaction matrix between sub-catchments based on gravity model.

| No. | 1 | 2 | 3 | 4 | 5 | 6 | 7 | 8 | 9 | 166 |
|------|------|------|------|------|------|------|------|------|------|------|
| 1 | 0.48 | 0.32 | 0.37 | 0.37 | 1.09 | 0.27 | 0.26 | 0.26 | 0.32 | 1.15 |
| 2 | 0 | 58.43 | 19.58 | 9.23 | 19.26 | 4.04 | 4.98 | 2.83 | 149.97 | 0.90 |
| 3 | | 0 | 51.95 | 12.50 | 21.11 | 4.32 | 6.57 | 2.72 | 23.05 | 0.67 |
| 4 | | | 0 | 41.79 | 45.52 | 10.19 | 20.02 | 4.49 | 9.15 | 0.86 |
| 5 | | | | 0 | 267.51 | 43.78 | 95.69 | 12.07 | 4.47 | 1.02 |
| 6 | | | | | 0 | 955.43 | 83.83 | 112.36 | 9.28 | 3.61 |
| 7 | | | | | | 0 | 21.77 | 19.01 | 2.18 | 0.88 |
| 8 | | | | | | | 0 | 5.98 | 3.09 | 0.74 |
| 9 | | | | | | | | 0 | 1.43 | 1.06 |
| ... | | | | | | | | | | |
| 166 | | | | | | | | | | ... |
| | | | | | | | | | | ... |

### 3.4. Spatial Autocorrelation Analysis of Sub-Catchments

The potential runoff network density (d) in eight scenarios (0, 18.93, 30, 40, 50, 100, 300, and 500 of the gravity coefficient) was calculated (Figure 7). On this basis, a spatial autocorrelation analysis (Moran's I coefficient) of runoff density in sub-catchments was carried out. It was found that (1) in eight scenarios, Moran's I coefficient was greater than 0, and Z > 2.58; $p < 0.01$, indicating that the distribution of potential runoff density was significantly positively correlated with the distribution of catchments and showed a significant aggregation, with a gradually decreasing trend from the inner central district to the outer city edge. (2) With the increase in the coefficient of gravity, Moran's I presented a trend of first increasing and then decreasing. First, the spatial distribution of runoff density was positively correlated with the gravity coefficient. The sub-catchment areas with high gravity coefficients first generated runoff and showed high aggregation. Second, with the increase in runoff density in the region with a low gravity coefficient, the spatial distribution of runoff density in the whole study area was averaged, and the spatial aggregation decreased. This result not only reflected the spatial aggregation characteristics of runoff but also suggested that the potential surface runoff is mainly distributed in the urban center at the initial stage (Table 5).

This outcome indicated that the impermeable surface area, i.e., the number of buildings and streets in the city center, was higher than that in the surrounding areas, resulting in a high spatial dependence among sub-catchments, and the potential surface runoff connections were more likely to interact with each other. In general, the potential surface runoff density in the central city under the condition of different gravity coefficients was significantly higher than that outside the city center. Further observations showed that the distribution of potential runoff density revealed a significant multicore and multistage sequence distribution pattern, which is consistent with the characteristics of land change in the central urban area.

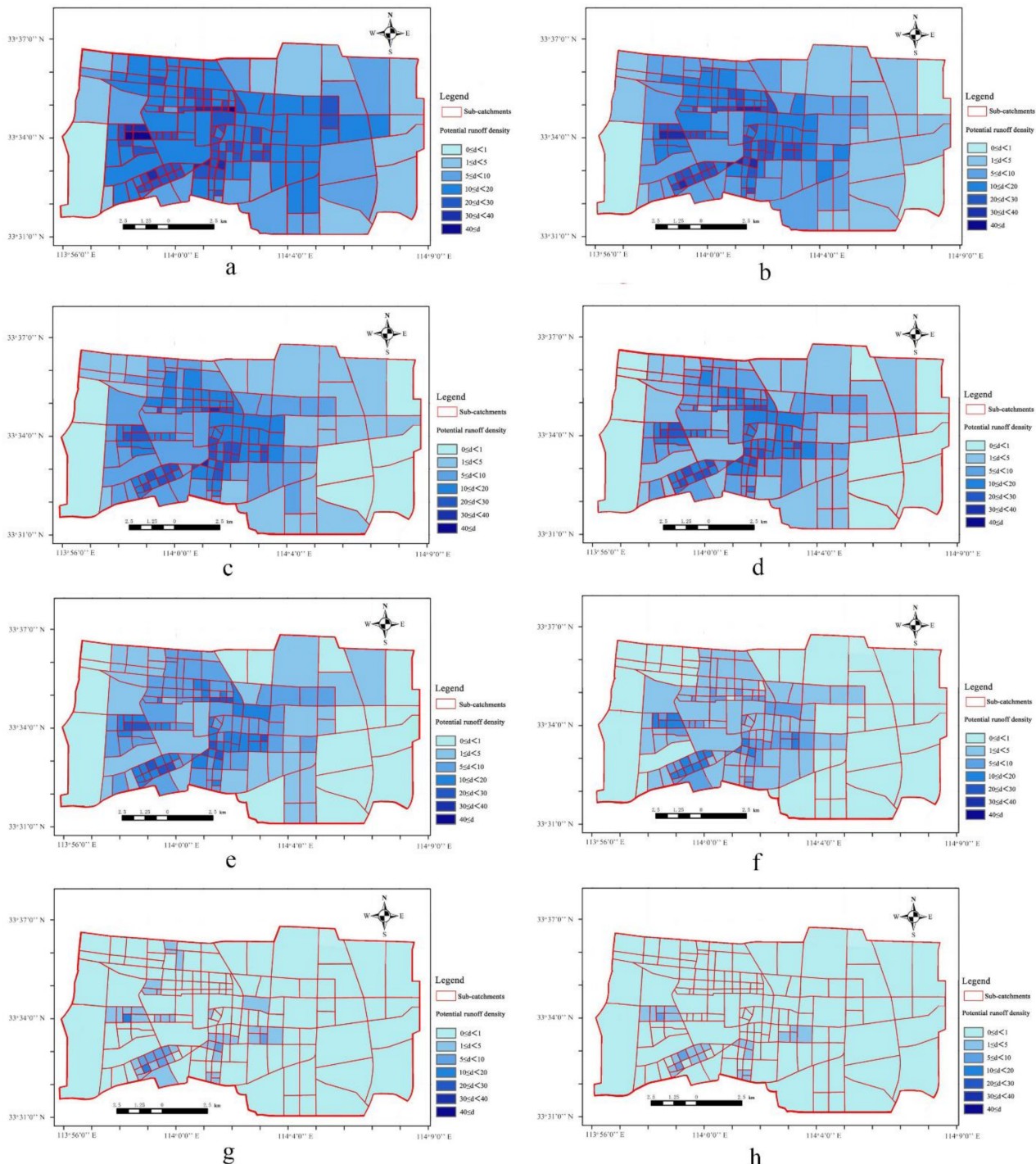

**Figure 7.** Potential runoff coefficient under the condition of different gravity density distribution of Luohe central district ((**a–h**) are the distribution state of potential runoff density when the gravity coefficients are 0, 18.93, 30, 40, 50, 100, 300, and 500, respectively).

**Table 5.** Analysis of potential surface runoff distribution characteristics.

| Gravity Matrix | 0 | 18.93 | 30 | 40 | 50 | 100 | 300 | 500 |
|---|---|---|---|---|---|---|---|---|
| d | 17.69 | 11.78 | 14.56 | 9.78 | 8.36 | 4.20 | 1.06 | 0.49 |
| Moran's I | 0.22 | 0.31 | 0.32 | 0.30 | 0.29 | 0.26 | 0.27 | 0.21 |
| Z | 12.08 | 16.92 | 17.49 | 16.46 | 15.66 | 14.28 | 15.25 | 12.40 |
| *p* | 0.00 | 0.00 | 0.00 | 0.00 | 0.00 | 0.00 | 0.00 | 0.00 |

## 4. Discussion

### 4.1. Advantages of Potential Surface Runoff Simulation

For urban areas, improving the conditions of surface runoff distribution and constructing land expansion are coexisting processes. The resulting orthophoto maps and DSM model was satisfactory and provided the sufficient details to define runoff paths based on elevation. This model was validated by Chi-square analysis of runoff along the urban surface landscape. This has reinforced the conclusions reached about the accuracy of high precision elevation data from Hoffmann et al. [42], Chigbu et al. [43], Sami et al. [17], Hasan [44] and my own research by SWMM [16]. It is worth mentioning that SWMM only requires the permeability coefficient of landscape layouts [45]. Further, our modeling methods consisting of the comprehensive surface landscape information, the orthophoto maps, and DSM's resolution were effective enough to obtain its spatial layouts, i.e., the slope, relief amplitude, roughness, and LUCC of study area. Studies have shown that runoff increases are caused by urban LUCC changes, and high runoff aggregation is mainly concentrated in the Luohe central district. Industrial areas, commercial districts, and high-density residential areas tend to have a greater impact on runoff. Some research results showed specific local relationships between runoff indices and landscape factors [46]. Controlling landscape pattern processes could change runoff distribution between temporal and spatial; when similar general cases reoccur between sub-catchment areas, similar spatial patterns result. Currently, the increase in impermeable land surfaces caused by urban expansion is an inevitable trend. How to construct urban landscape structure reasonably and effectively, including permeable and impermeable landscapes, will be the key point of urban rainwater resource allocation in the future.

The aggregation distribution of surface runoff in the Luohe central district was closely related to urbanization development. This phenomenon appeared to occur frequently in high-density areas [47]. The permeable landscape is the basis of GI and usually includes green space, farmland, woodland, and water, which can reduce infiltration excess (Hortonian) and the lower the rainfall-excess rates, the longer the surface runoff concentration time and the time of the peak, and the smaller the peak runoff value at any distance [48]. So, effectively improving the collection, transportation, utilization, and regeneration of stormwater requires not only the improvement of municipal pipe networks but also the reasonable and adequate utilization of landscape layouts. Studies of the overall surface landscape can reflect the functional strength of urban stormwater carrying capacity. Surface runoff is significantly influenced by its landscape layout—permeable and impermeable. These effects were treated through the landscape resistance [49]. The effects of the resistance on surface runoff could be negative or positive [50]. With the help of "source" to "sink" identification (from a sub-catchment to others) and resistance surface (MCR model), the runoff transmission paths could be analyzed [51,52]. Thus, the MCR model was employed to analyze the distribution of surface runoff. Through statistical analysis, it was found that the capacity of the Luohe green space to deal with runoff was obviously insufficient; green space accounts for 14.64% of the studied district, but only 5.65% of runoff goes through the green spaces. The length of runoff in green spaces accounts for 7.51% of the total runoff. Regarding the hydrologic response, accurate information regarding the position and layout of the landscape structures was needed to simulate its geometry of surface runoff [17]. The potential runoff spread in different scale of sub-catchments was investigated. Although such data could be easily gathered in small areas, this task could be time-consuming for larger study sites, remaining as a challenge if upscaling of the results is planned [53]. Nevertheless, these methods require a high-resolution surface topography data and satellite images [54]. In our case, it was shown to be accurate enough to explain the relation between landscape layout and runoff distribution. By simulating the distribution of potential runoff, the key areas to be transformed can be identified, and the diversion design can be carried out at the intersection of surface runoff [55]. It is important to take full advantage of the existing permeable landscape. The simulation results can accurately locate the inlet and outlet of green space, guide the transformation of microtopography or landscape, guide

the surface runoff into the permeable surface landscape, and increase the utilization rate of permeable green space. Thus, simulating the local hydrological effect of landscape layout could be an important tool used by managers to improve their management efficiency of such measures and for local promotion of their benefit [2,56].

*4.2. Strategies for the Protection of Suitable Distribution of Urban Stormwater*

The organic combination of urban permeable landscapes and impermeable landscapes plays an important role in the stormwater security of urban areas [57]. Different landscapes have different resistances to runoff, and different sub-catchments have different gravity impacts on runoff. Therefore, the values and map of the potential runoff distribution of Luohe provides guidance for urban landscape planning. There are a number of measures that could be considered for disaster reduction [58]. Setting up protection green corridors along the urban trunk road and branch road would be beneficial for organic evacuation of surface runoff. Guided spaces and protected green belts around the urban area next to the river could help prevent the spread of urban flood pollution. Establishment of rain gardens, bio-retention, and permeable areas combined with existing natural resources and even gardens on roofs, which account for 20.09% of the total area of the Luohe central district would improve the quality of life for urban residents and protect the environment. In addition, the organic combination of urban surface landscapes plays an important role in urban security and maintenance of urban biodiversity. Luohe construction continues to grow, showing spatial agglomeration. The central urban area has the most direct impact on the distribution of surface runoff, with the most significant feedback effect. The distribution of surface runoff in Luohe was consistent with the expansion trend of urban construction. Currently, the permeable surface landscape of Luohe city, especially the green space, is almost entirely dependent on the new development of communities and buildings, so in the LUCC map, it is difficult to observe a relatively continuous greenway or green belt. The simulation results can accurately locate the inlet and outlet of green space, guide the transformation of microtopography or landscape, guide the surface runoff into the permeable surface landscape, and increase the utilization rate of permeable green space. Establishing greenbelts on both sides of roads and rivers within the region and strengthening the functional connection of permeable space through integrative green space could increase infiltration, retention, and purification processes of runoff. Building a hydrologic park suitable for public events would increase the recreational functions of permeable space. Removing unnecessary urban structures in central district with connected runoff channels could guide organic evacuation of runoff from urban spaces and avoid growth in submerged areas and low land-use efficiency [59].

**5. Conclusions**

The MCR model was used to identify the runoff path from the surface landscape, and we compared the potential runoff between different sub-catchments using the gravity model and spatial autocorrelation from landscape properties in Luohe.

Our results showed that total permeable land (including green space, farmland, and unused land) in the central urban area accounts for 60.98% of the city's total land area, and green space occupies only 14.64%. There is a large and growing area of construction land. To meet requirements for rapid urbanization and to protect suitable ecological areas and green spaces, a reasonable allocation of both areas and a suitable spatial pattern of the two LUCCs are important.

For suitable surface runoff distribution, time must be taken to establish a unified planning of the city's surface landscape. Permeable landscape in urban areas and protection is primary, but still needs scientific urban planning to ensure landscape connectivity and the spatial integrity of critical land. Suitable construction land and rational development are needed along with increasing the infiltration efficiency of land use.

Finally, HD remote sensing was used to obtain surface landscape data and is an efficient method. However, due to the error of LUCC accuracy, the image is still different

from reality; selection factors and weight allocation of the resistance surface in the MCR model revealed some subjective elements, and selection factors were not comprehensive enough because of the limitations of information and data collection in Luohe. Therefore, in future studies, it is necessary to improve the experiment and investigate more parameters.

**Author Contributions:** T.B. and Y.W. conceived of the presented idea and wrote the manuscript, with support from K.B. and J.Z. helped supervise the manuscript. All authors discussed the results and contributed to the final manuscript. All authors have read and agreed to the published version of the manuscript.

**Funding:** This research was funded by statutory funds of faculty of College of Horticulture and Landscape of Yunnan Agricultural University, Yunnan Province Education Department, China, grant number 2020J0274.

**Institutional Review Board Statement:** Not applicable.

**Informed Consent Statement:** Not applicable.

**Data Availability Statement:** The data (includes charts and raw data) supporting reported results can be found at https://orcid.org/0000-0001-9647-4646 (accessed on 1 May 2021).

**Acknowledgments:** We thank the Luohe City Municipal Engineering Design Institute and College of Forestry, Henan Agricultural University for urban survey original data. We thank Huaibi Zhang, Plant and Food Research, New Zealand for his contribution to the revision of this paper. We also thank Audrey L. Mayer (MTU) and Emily S. Minor (IESP) offered suggestions for modification. AndXinyu Wang and Xiaoyan Wang, for assistance with classifying orthographic imagery. Although the study involved the distribution mode of rainwater runoff in urban spaces and has positive significance for urban construction, it has not been implemented in actual construction and is still in the research stage. Consequently, the views, interpretations, and conclusions expressed in this article are solely those of the authors and do not necessarily reflect or represent local government views or policies.

**Conflicts of Interest:** The authors declare no conflict of interest. The funders and material provider had no role in the design of the study; in the collection, analyses, or interpretation of data; in the writing of the manuscript, and in the decision to publish the results.

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
