# Peer review of "Highly Resolved Runoff Path Simulation Based on Urban Surface Landscape Layout for Sub-Catchment Scale"

_water, doi:10.3390/w13101345_

Round 1

Reviewer 1 Report

excellent paper

I found very well done and developed

Just minor comments to improve the discussion and introduction with a more  holistic and general overview of the research done

Author Response

Thank you for your advice. We would like to thank for review of our manuscript. We have followed all suggestions and we believe this will improve our manuscript significantly. Moreover, the English language has been corrected by Journal English proof editing system.

Please find below the response to the suggestion:

  1. To improve the discussion and introduction with a more  holistic and general overview of the research done
  2. we study the paper you recommended and applied it in the paper, and modified the content in detail

We have include some more information in the introduction and discussion chapter.

Reviewer 2 Report

The manuscript entitled “Highly Resolved Runoff Path Simulation base on Urban Surface Landscape Layout for Subcatchment Scale”, by T. Bai, K. Borowiak, Y. Wu & J. Zhang, presents an interesting work.

In general, the manuscript should be acceptable for publication but some serious problems must be repaired prior to publication. It needs some significant improvement. Some suggestions are as follows:

  1. Please use different terms in the “Title” and the “Keywords”.
  2. The English language usage should be checked by a fluent English speaker. It is suggested to the authors to take the assistance of someone with English as mother tongue.
  3. The abstract should state briefly the purpose of the research, the principal results and major conclusions. An abstract is often presented separately from the article, so it must be able to stand alone.
  4. In all maps you could put coordinates.
  5. Correct references in the text and the reference list according to the journal’s format. Please format the references’ list by using the correct journal abbreviations. See the following link: https://images.webofknowledge.com/images/help/WOS/A_abrvjt.html
  6. Please be careful with the spaces between the words.
  7. You could enrich the scientific literature.
  8. Please justify convincingly why this manuscript (method, thematology etc) connected with Water’s content and scope. Perhaps the using of proper literature would be helpful. Eg:

- Cai, Q.-C.; Hsu, T.-H.; Lin, J.-Y. Using the General Regression Neural Network Method to Calibrate the Parameters of a Sub-Catchment. Water 2021, 13, 1089. https://doi.org/10.3390/w13081089

- Đukić, V.; Erić, R. SHETRAN and HEC HMS Model Evaluation for Runoff and Soil Moisture Simulation in the Jičinka River Catchment (Czech Republic). Water 2021, 13, 872. https://doi.org/10.3390/w13060872

- etc

9. The authors could take into account the following publications:

- Bathrellos GD, Karymbalis E, Skilodimou HD, Gaki-Papanastassiou K, Baltas EA (2016): Urban flood hazard assessment in the basin of Athens Metropolitan city, Greece. Environ Earth Sci, 75 (4): 319.

Author Response

We would like to thank for review of our manuscript. We have followed all suggestions and we believe this will improve our manuscript significantly. Moreover, the English language has been corrected by Journal English proof editing system.

Round 2

Reviewer 2 Report

About the Manuscript I recommend to be accepted in the present form.